

# Modular Assessment of Rainfall–Runoff Models Toolbox (MARRMoT) v2.1: an object-oriented implementation of 47 of your favourite hydrologic models for improved speed and readability

Luca Trotter[1], Wouter J. M. Knoben[2], Keirnan J. A. Fowler[1], Margarita Saft[1], Murray C. Peel[1]

[1]Department of Infrastructure Engineering, University of Melbourne, Melbourne, Parkville VIC 3052, Australia
[2]Centre for Hydrology, University of Saskatchewan, Canmore, Alberta T1W 3G1, Canada

*Correspondence to*: Luca Trotter (l.trotter@unimelb.edu.au)

**Abstract.** The Modular Assessment of Rainfall–Runoff Models Toolbox (MARRMoT) is a flexible modelling framework reproducing the behaviour of 47 established hydrological models. MARRMoT can be used to calibrate and run models in a

user-friendly and consistent way and is designed to facilitate the sharing of model code for reproducibility and to support intercomparison between hydrological models. Additionally, it allows users to create or modify models using components of existing ones. We present a new MARRMoT release (v2.1) designed for improved speed and ease of use. Whereas improved computational efficiency was the main driver for this redevelopment, MARRMoT v2.1 also succeeds in drastically reducing the verbosity and repetitiveness of the code, which improves readability and facilitates debugging. The process to create new

models or modify existing ones within the toolbox is also simplified in this version, making MARRMoT v2.1 accessible for researchers and practitioners at all levels of expertise. These improvements were achieved by implementing an object-oriented structure and aggregating all common model operations into a single class definition that all models inherit from. The new modelling framework maintains and improves on several of the good practices built into the original MARRMoT and includes a number of new features such as the possibility of retrieving more output in different formats, which simplifies

troubleshooting, and a new functionality that simplifies the calibration process. We compare outputs of 36 of the models in the framework to an earlier published analysis and demonstrate that MARRMoT v2.1 is highly consistent with the previous version of MARRMoT (v1.4), while achieving a 3.6-fold improvement in runtime on average. The new version of the toolbox and user manual, including several workflow examples for common application, are available from GitHub (https://github.com/wknoben/MARRMoT, last accessed 12/5/2022. DOI: 10.5281/zenodo.6484372).

# 1 Introduction

The Modular Assessment of Rainfall-Runoff Models Toolbox (MARRMoT) is a flexible modelling framework which reproduces components and behaviours of established conceptual hydrologic models, while allowing for their modification and reshuffling (Knoben et al., 2019). MARRMoT implements 47 conceptual hydrologic models in MATLAB in a consistent way. The models coded in the framework are all established conceptual hydrologic models commonly used in academia and





industry, they include GR4J (Perrin et al., 2003), Sacramento (Burnash, 1995), HBV (Lindström et al., 1997), VIC (Liang et al., 1994) and TOPMODEL (Beven and Kirkby, 1979), amongst others. The framework supports research reproducibility by making it easy to access and share versioned model code. Additionally, MARRMoT's consistent implementation of these models facilitates intercomparison studies, encouraging a more active approach to model evaluation and selection for specific applications, hence reducing researcher bias (Herath et al., 2020; Peel and McMahon, 2020). The possibility to easily modify

individual model components and routines, both from scratch or by substituting in components from other models, encourages the use of models for hypotheses testing (Clark et al., 2011) and simplifies processes for model diagnosis and improvement (e.g. Westra et al., 2014). On top of fostering a culture of innovation and experimentation with regards to hydrologic model development and application, flexible frameworks such as MARRMoT provide easy-to-use tools with built-in mechanisms that ensure the use of good and consistent modelling practices and numerical approximation schemes, hence lowering the

threshold for creating new, good quality model structures and modifying existing ones.

MARRMoT follows a number of good practices for model development when encoding hydrologic models within the framework (Knoben et al., 2019). Specifically, MARRMoT models are defined in terms of their constituting ordinary differential equations (ODEs) in state-space formulation and their definition is separate from the numerical methods used to solve them, which allows for clearer identification of model processes and behaviour and differentiation of sources of errors,

while facilitating parameter estimation (Clark and Kavetski, 2010; Kavetski and Clark, 2010). When it comes to solving ODEs, MARRMoT uses a fixed-step implicit Euler scheme as default, which, compared to explicit schemes, provides more accurate and stable estimations (Kavetski et al., 2006). Numerical stability is also enhanced by using mathematical smoothing (through a logistic function) of storage and temperature thresholds in the constitutive equations; this also contributes to better parameter estimation (Kavetski and Kuczera, 2007). Finally, whenever possible, MARRMoT avoids operator-splitting numerical

approximations by solving all model ODEs at once, hence sidestepping the need to make assumptions on the order of model fluxes which could prove physically inaccurate (Fenicia et al., 2011).

In its current format, MARRMoT prioritises readability and implementing robust mathematical approaches over speed (Knoben et al., 2019) and feedback from the users suggests runtime to be the largest obstacle to widespread use of the framework. With the principal aim of improving runtime efficiency, we radically restructured MARRMoT following the tenets

of object-oriented programming (see Section 2 below). Compared to the previously available versions (MARRMoT v1), the version presented here (MARRMoT v2.1) simultaneously achieves better speed, readability and user-friendliness. Additionally, it drastically reduces the verbosity and repetitiveness of the code, and hence the possibility of errors due to typos; it allows for simultaneous solving of model equations in all cases, including when routing functions operate in-between stores; it provides greater control on model outputs with enhanced capabilities for debugging and error detection; and it lowers the

threshold for implementing new (or modifying existing) model structures.

While working on this update, care was taken to ensure the maximum possible compatibility with previous versions of MARRMoT (v1) in order to facilitate transition as much as possible for existing MARRMoT users. In this regard, inputs and outputs remain identical in this version of MARRMoT to ensure reusability of any pre- or post-processing routines; whereas



commands to run simulations and calibrate models changed minimally. Details on minimum requirements to update
MARRMoT v1 code for v2 are given in the user manual (Section 2.5) included in the repository. MARRMoT v2.1 runs in
Octave as well as MATLAB.

## 2 Technical improvements

A schematic overview of the structure of the new object-oriented MARRMoT v2.1 framework is shown in Figure 1. Compared
to MARRMoT v1, where models were defined as functions (see Knoben et al., 2019 Figure 1), each model in the framework
is here conceptualised as an *object class*. In object-oriented programming, a class is a descriptor of similar objects (Stefik and
Bobrow, 1985); classes are defined in terms of state variables (known as attributes or properties) and behaviours (or methods)
and can be thought of as templates or placeholders that may be populated in different ways depending on context. Objects, on
the other hand, are instances or realisations of the class they belong to (Stefik and Bobrow, 1985). Crucially, within an object-
oriented project, it is possible to define trees or lattices of classes which allow objects to inherit not only the attributes and
methods defined in their own class, but also all those defined in each superclass (or parent class) that their (sub)class is a child
of (Stefik and Bobrow, 1985). As an example, consider a conceptualisation of your house pets Fido the dog and Wanda the
fish: Fido and Wanda are objects, or instances of their classes, dog and fish respectively. The dog class would define dog-
specific methods such as fetch, whereas the fish class might include a swim method; additionally, both classes would have a
superclass such as animal as their parent, which would contain definitions of methods common to all animals, such as feed or
sleep; in this way Fido would inherit methods fetch, feed and sleep, whereas Wanda's methods would include swim instead of
fetch, but still include feed and sleep from its parent class animal.

In MARRMoT v2.1, we have defined a simple two-level class tree. At the top level is the superclass known as
*MARRMoT_model*, which defines all operations that are common to all models (see Section 3.1.1 of the user manual for a list
of these common methods). Each of the 47 individual model structures is then defined as its own child class inheriting all
common operations from the superclass and with additional methods defined to match the specific model formulation.
Instances of these classes, the model objects, act as containers for the data and procedures needed to run the framework on
specific case studies. The user takes these model objects, populates them with parameters, initial state variables and other
necessary inputs and uses them to run model simulations.

A superclass defines all common methods, this centralises input checks, all procedures to solve equations and produce outputs.

Each hydrologic model is a defined as a (sub)class of the *MARRMoT_model* superclass. Here, all model-specific equations are defined.

*MARRMoT_model* (superclass)

Attributes

All *attributes* are declared here as empty templates, and populated either in individual model classes or by user input.

Methods

1. Deal with inputs
2. Create ODEs approximations and solve them
3. Run simulation
4. Help with model calibration
5. Other/helper functions

*model_1* (class)

*model_2* (class)

...

*model_m* (class)

Inherits all *attributes* and *methods* from the superclass

Additional methods

1. Set model- and parameter-specific attributes
2. Define model equations in state-space formulation
3. Define model-specific actions to run at t=0 and at the end of each time-step

*flux_1*

*flux_2*

...

*flux_n*

(function)

Model constitutive equations are unique sets and arrangements of fluxes

MARRMoT

Specific model functions are called by the methods from the superclass

Modelling study

Climate observations (P, PET, T)

Initial store values

Time-stepping and solver settings

Model parameter set

Inputs and settings are stored in the model object as attributes

**Model object**

A model object is an instance of a model class. Model objects are created by the user, populated with attributes and used to run simulations.

After a model object runs a simulation, any of its attributes can be retrieved as outputs.

*Observations*

*Simulation output:*

Flow, fluxes, stores, water balance…


**Figure 1: Schematic overview of the MARRMoT framework in it v2.1 implementation. A detailed description of the object-oriented MARRMoT implementation can be found in Sections 2.1 and 2.2.**





## 2.1 The *MARRMoT_model* superclass

The creation and definition of a superclass was motivated and guided by the observation that all models in the framework share
many common operations, such as a those handling meteorological inputs, defining numerical solver settings, and generating
outputs. Consolidating all of these in a common location has major advantages as it significantly simplifies the individual
model files (i.e. the class definition files of each individual model). This enhances readability and facilitates debugging when
running modelling studied, additionally, it lowers the threshold for implementing new model structures and reduces the risk
of typos or copy-paste errors. Additionally, it simplifies and streamlines the process of deploying and testing changes (e.g. to
the equation solving routine or the output format) across all models, by simply modifying the *MARRMoT_model* definition
file, which all model classes inherit from.

In terms of attributes and methods, in the superclass we declare all model attributes and define a number of common model
methods. Attributes are only declared as empty templates in memory and no attribute is populated (i.e. assigned a specific
value) directly in the superclass definition file. For example, the superclass declares that all models have attributes called
*numParams*, *numStores* and *theta* to store the number of model parameters, number of stores and a set of parameters
respectively, but the values of these properties, being model- or simulation-specific, are assigned later. We distinguish model
attributes in three groups. Firstly, static attributes, which are model-specific and common to all instances of the same model
(e.g. number of parameters or stores). Secondly, user-defined attributes, which are simulation-specific and static throughout
a simulation; they are populated from input by the user directly (e.g. set of parameters, initial store values, climate data) or
inference from the specified parameter set (e.g. store maxima and minima). Finally, dynamic attributes, which are populated
and modified internally throughout a model run (e.g. store or flux values at every timestep).

In addition, methods to perform all common model operations are defined in the superclass. On top of helper methods to check
and manage user input formats or specify default options and settings, model methods perform three key operations: create
approximation of models' ordinary differential equations (ODEs) from model-specific constitutive equations and numerically
solve them (Section 2.1.1); step through a simulation and return its output to the user (Section 2.1.2); and, newly introduced
in this version of MARRMoT, calibrate a model to a set of observations (Section 2.1.3). The following paragraphs provide
further details on each of these operations, focusing on functional differences from MARRMoT v1 (Knoben et al., 2019).

### 2.1.1 Numerical ODEs approximation and solving

As mentioned in the Introduction, the principal aim for the development of MARRMoT v2.1 was to improve runtime
efficiency. This was achieved by modifying the way model ODE approximations are solved (see Section 3.2 and
). In a sense, the new object-oriented structure is actually a by-product of this modification. With the new structure, we
concentrated the solving routine for all models into a single place, which allows changing and experimenting with the solving
routine much easier to deploy across all models, without the need to modify each individual model files.





As with previous versions, all models' constitutive ODEs are approximated in MARRMoT v2.1 using a fixed-step implicit
Euler numerical scheme (following suggestion by Clark and Kavetski, 2010). However, the details of how this is solved vary
from the previous MARRMoT version. There, MATLAB's proprietary root-finding functions *fsolve* and *lsqnonlin* were used:
the former as a first attempt and, if a suitable solution is not found within the specified tolerance, the latter as a more robust,
but slower alternative. In MARRMoT v2.1, the solution is initially attempted using an open source implementation of the
Newton-Raphson algorithm enhanced by line searches (also following Clark and Kavetski, 2010). In most cases, the Newton-
Raphson solver used in MARRMoT v2.1 is sufficient to identify a suitable solution; when this doesn't happen, the framework
reverts to the same *fsolve* and *lsqnonlin* functions used in the previous versions. A solution is deemed suitable if the norm of
its residuals is below a pre-determined acceptance threshold, passed by the used together with other necessary options (see
Section 2.2 of the user manual).

The object structure also easily allows keeping a log of the operations of the equation solving routine as an attribute to the
model object. For every step, this includes solver used, value of the error on the ODEs' approximation and number of iterations
needed to reach the solution. This can be retrieved after a simulation to check the quality of the solutions and adjust settings
and parameters if needed.

Finally, there were certain circumstances where the previous versions were programmed not to attempt concurrent solving of
equations, and this has been improved in the updated version. Specifically, in MARRMoT v1, when a model had fluxes routed
through unit hydrographs (UH) in-between stores, the solver would solve stores up- and downstream of the UH separately,
effectively using a form of operator-splitting (OS). OS approaches have a number of limitations and can incur numerical errors
due to the physically unrealistic assumption that processes in hydrological systems operate in a predetermined order (Fenicia
et al., 2011). Whereas the structure of the previous MARRMoT version made this form of OS particularly complicated to
remove, the new object-oriented structure allowed us to more easily code MARRMoT v2.1 to solve model equations
simultaneously in all cases, and this is set as the default method in the new version.

### 2.1.2 Simulation and output retrieval

The syntax to run a model simulation and the format of the outputs are kept as consistent as possible to those of MARRMoT
v1 to facilitate transition to the new version for users already familiar with the framework. However, the new implementation
is more flexible, allowing to both run a simulation and retrieve its outputs in a variety of ways to match different workflows
(see the Sections 2.2 and 2.3 of user manual for details). Additionally, the object-oriented structure of MARRMoT v2.1 allows
for all outputs and information about a model run to be stored as attributes of the model object itself at the end of a simulation.
These include:

- values of all stores and fluxes at all timesteps;
- information about the operation of the numerical solver used and the quality of the solution found (as mentioned
above);
- a copy of the parameter set, initial store values and the climate data used to force the model;





- the settings and tolerances set for the numerical solver; and
- the final state of the UHs, containing values of fluxes that are still to be routed at the end of the simulation.

Storing outputs as attributes not only makes them easier to retrieve after a simulation, but also allows the user to save the
model object that contains this information for later retrieval.

### 2.1.3 Model calibration

The *MARRMoT_model* superclass contains a default calibration method to help fit a model to a set of observations. Whereas
MARRMoT's outputs can be fed into any external optimisation algorithm for calibration (like users would do with MARRMoT
v1), the *calibrate* method simplifies the calibration process. To use the method, the user needs to define what objective function
and optimiser should be used for the calibration process. Whereas these can take any functional form, they must have the
correct input-output format. For the objective function, a number of commonly used objective functions are implemented in
the MARRMoT repository (already in version v1) for the user to choose or to use as templates to create their own function.
For the optimisers, the calibration method expects this to have the same input-output format as MATLAB's proprietary
optimisers (e.g. *fminsearch*) which can therefore be used directly within the method. Additionally, the MARRMoT v2.1
repository also contains an implementation of the Covariance Matrix Adaptation Evolution Strategy (CMA-ES) algorithm
(Hansen and Ostermeier, 1996; Hansen et al., 2003) that matches the expected format and is ready to be used with the *calibrate*
method within MARRMoT v2.1. CMA-ES is widely used in a variety of fields (Hansen, 2009) and it was shown to perform
favourably in hydrological model calibration compared to other algorithms (Arsenault et al., 2014).

### 2.2 Individual model classes

Given that the guiding principle when defining the superclass was to include in it everything that is shared between all models,
when writing model-definition classes or model files, the goal was to reduce these to their bare minimum, only including in
them the definition of each individual model's structure. As already mentioned, each model class is a child of the
*MARRMoT_model* superclass from which it inherits all attribute and methods described above.

To simplify the process of creating new and/or modifying existing model classes, these all have the same structure, and all the
model-specific information is contained in four methods:

1. **Creator method**: The creator method runs every time a model object is created and populates all static model attributes (e.g. number and names of stores and fluxes, parameter ranges).
2. **Initialiser:** The initialiser runs once at the beginning of every simulation, to set up the model run, for example by calculating store maxima and minima from parameters, initialising unit hydrographs or calculating additional derived
185        parameters.
3. **Within-timestep calculations:** The functions defining the inner functioning of the model are coded in state-space formulation as an additional method which is called by the solver method from the superclass at every timestep to solve the model's ODEs.





4. **Between-timestep updating:** The stepping method which runs at the end of every timestep and is primarily used to update unit hydrographs and other routing mechanisms.

Compared to the way individual models are coded in MARRMoT v1, this structure substantially reduces opportunities for typing errors by reducing the verbosity of the code (e.g. in MARRMoT v1 model constitutive equations were repeated in each model file at least twice, whereas now, they only need to be coded once). Additionally, structuring the model definition into four well-defined methods makes the inner structure and functioning of the models clearer, hence making it easier for unfamiliar users to create new models and modify existing ones. The definition of model equations as a dedicated method in the model file provides a clear separation between ODEs definition and their approximation and solving, which happens at the superclass level.

## 2.3 Other changes

MARRMoT v2.1 contains additional changes to the structure and form of helper functions to ensure that they work efficiently with the new model structure. Compared to MARRMoT v1, flux files, i.e. the files defining the form of individual fluxes that make up the model constitutive equations, have been changed from anonymous functions to regular functions, without loss of readability thanks to the object oriented structure. Unit-hydrograph functions have also been simplified in their form and structure without affecting their functionality. Additionally, two helper functions *route* and *update_uh* are now used to simplify the use of unit hydrographs when creating new models or modifying the existing ones, substantially improving readability and ease of use. Finally, the code of the example objective functions was modified to allow the user to set what timesteps to calculate the fitness on; using this function together with the calibration method allows to set up warm-up periods and specify what periods (even non-contiguously) to use to for calibrating the models.

## 3  Test cases

We compare the performance of MARRMoT v2.1 against MARRMoT v1.4, which is the next-most-recent version of MARRMoT published in the MARRMoT GitHub repository. In order to evaluate the consistency of the simulations of MARRMoT v2.1 to the previous version, we use an intermediate version (MARRMoT v2.0) which implements the object-oriented structure described above, but maintains the same equation solving routine as MARRMoT v1.4. This allows us to distinguish the effects of the change in structure from those of the new root-solving scheme. Table 1 summarises the differences between the versions of MARRMoT. Versions 1.4, 2.0 and 2.1 are used for the application test.  Note: even though Versions 1.0-1.4 are outside of the scope of the MARRMoT updates being reported in this article, we feel it is a useful resource for users to have a table that gives full details of the different versions, which is why all details (including bug fixes, etc.) are reported in Table 1 regardless of the version they relate to.



**Table 1: Summary of differences between the versions of MARRMoT used for the application test.**

| Version | Change from previous version | DOI |
|---------|------------------------------|-----|
| v1.0 | MARRMoT submitted for peer-review | 10.5281/zenodo.2482542 |
| v1.1 | Peer-reviewed MARRMoT (Knoben et al., 2019). Bugs fixed: typo in logisitic smoothing function. Added pure time delay Unit Hydrograph to m05. | 10.5281/zenodo.2677728 |
| v1.2 | Added parameter range display during runs | 10.5281/zenodo.3235664 |
| v1.3 | Bugs fixed: added missing constraint in interflow_9 flux | 10.5281/zenodo.3552961 |
| v1.4 | Bugs fixed: timestep size in water balance calculation; timestep size of certain fluxes used by m05, m15, m37, m44; sign error in m09 and m07 model function; arguments to evap_16() in m17, m25; input to saturation_1() in m30, m31, m32, m34; missing flux in m36. Updated workflow example 4 to work with Octave. Added model m47. Added several efficiency metrics and ability to specify warm-up period in metric calculation. Reduced numerical instabilities in m37. | 10.5281/zenodo.6460624 |
| v2.0 | Object-oriented structure | 10.5281/zenodo.6483914 |
| v2.1 | New root-finding routine using Newton-Raphson solver | 10.5281/zenodo.6484372 |

### 3.1 Methodology for test cases

We use data from the 559 catchments in the CAMELS data set (Addor et al., 2017) already used to calibrate and test 36 of the models in MARRMoT v1.0 (Knoben et al., 2020). Here we test the same 36 models in MARRMoT versions 1.4, 2.0 and 2.1 (Table 1) and use the parameter sets calibrated by Knoben et al. (2020) as well as all MARRMoT settings specified in that
paper. In order to ensure that the new object-oriented structure does not modify the internal workings of the models, we compare the outputs from v1.4 and v2.0 and look at all model fluxes (internal and external) and stores at every timestep. Additionally, the effect of the new root-finding routine is evaluated by comparing the outputs of v2.0 to v2.1. Since the first test ensured that the internal processes are not altered, in the second test it is sufficient to look at the streamflow leaving the model to ensure consistency.

We use the Nash-Sutcliffe efficiency (NSE, Nash and Sutcliffe, 1970) to measure consistency of outcomes, where the timeseries produced by the version of MARRMoT with the lower version number is always used as the "true" timeseries. In order to avoid the large drops in NSE that may occur with even very small absolute differences in fluxes when the values of the "true" timeseries are very close to zero, we perturb both timeseries by the same random sequence of values in the order of $1 \times 10^{-6} mm$, which effectively makes any difference smaller than this value irrelevant.





All simulations are run as specified by Knoben et al. (2020): models are warmed-up to stabilise the stores by forcing them repeatedly with data for the year 1989 for a pre-set number of times (specified by Knoben et al. (2020) as the number necessary for the store values to stabilize within a prescribed tolerance and using 50 iterations as the pre-set maximum) and the simulation itself is run from 1 January 1989 to 31 December 2009, at daily temporal resolution. To compare runtimes, all simulations are run individually on a single core from an Intel(R) Xeon(R) Gold 6254 CPU @ 3.10GHz reserved for the purpose.

**3.2 Results**

Out of the 36 models tested, 30 returned the exact same output in their v2.0 implementation as they did in their v1.4 version for all fluxes and stores at all timesteps (within MATLAB's default precision of $1 \times 10^{-16}$). For the remaining six models, the absolute values of the difference in the annual water balance for all fluxes between the two versions are shown in Figure 2. These are never higher than $0.29 \ mm/year$ and in the greatest majority of cases several orders of magnitude lower. Note

that the four models where the differences are relatively higher (*m_07_gr4j*, *m_21_flexb*, *m_26_flexi* and *m_34_flexis*) contain a routing function in-between stores and their equations are therefore solved in two steps in MARRMoT v1.4 and in only one step in MARRMoT v2.0. This may introduce errors in v1.4 that are not present when all stores are solved simultaneously in v2.0. The differences in annual water balance of the remaining two models (*m_14_topmodel* and *m_27_tank*) never exceed $4.64 \times 10^{-7} mm/year$. Discrepancies in daily storage values have similar orders of magnitude (see Figures S1-S6).







**Figure 2: Differences in the annual water balance of simulations from MARRMoT v1.4 and v2.0. All models not shown have differences smaller than $1 \times 10^{-16} mm$. Cumulative distributions are calculated across all years and catchments aggregated.**


With high confidence that the change to an object-oriented structure does not modify the way models operate, we compare the outputs of v2.1 and v2.0 to evaluate the effect of the change in root-finding routine. In contrast to v1, both of these versions allow retrieval of information about the quality of the ODE solution found at every timestep. Figure 3 shows the number of simulations where models in each of these two versions of MARRMoT were not able to find a solution within the specified

convergence threshold ($0.1\ mm$) for at least one timestep; this means that none of the solvers (*fsolve*, *lsqnonlin* and Newton-Raphson, if applicable) was able to find a solution. In this scenario the simulation accepts the best solution found and continues. The 27 models not shown here solved the ODEs within tolerance for all timesteps and all catchments in both MARRMoT versions. As the figure highlights, the new root-finding routine found acceptable solutions in more catchments than the old solver for all models except for two (*m_02_wetland* and *m_37_hbv*). Note that the simulations where MARRMoT v2.0 (i.e.

the old solver) was not able to find acceptable solutions at all timesteps were not further considered when assessing the consistency of outputs with the new solver.



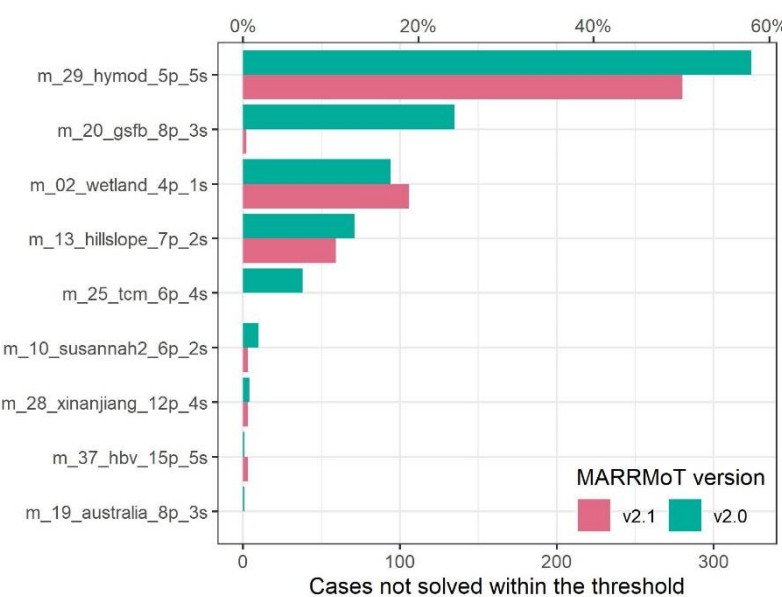

**Figure 3: Number of cases (out of 559) where at least one timestep is not solved within the convergence threshold for MARRMoT v2.0 and v2.1. All models not shown solved all timesteps of all simulations for both versions of MARRMoT.**

The final step to ensure that the output of MARRMoT v2.1 is consistent with previous versions of MARRMoT is the comparison of simulated streamflow between v2.1 and v2.0 with the NSE metric. For this comparison 30 of the 36 models achieved NSE > 0.999 for all 559 simulations. NSE values for the remaining six models are shown in Figure 4. As shown in the figure, all models have NSEs of at least 0.975 for all simulations, which indicates a very high level of consistency between the old and the new root-finding routines. The only exception to this is *m_17_penman* where simulations for 28 catchments

(5% of cases) have NSEs below this value and as low as 0.269. As indicated by the absence of *m_17_penman* from Figure 3, both versions of the model solve all timesteps satisfactorily for all simulations, suggesting that this model structure might be prone to issues of equifinality at the scale of the timestepping solver – that is, two or more solutions provide a satisfactory solution to the timestepping ODE implicit Euler scheme (as opposed to timeseries-wide calibration, which is the usual context of the word *equifinality*).





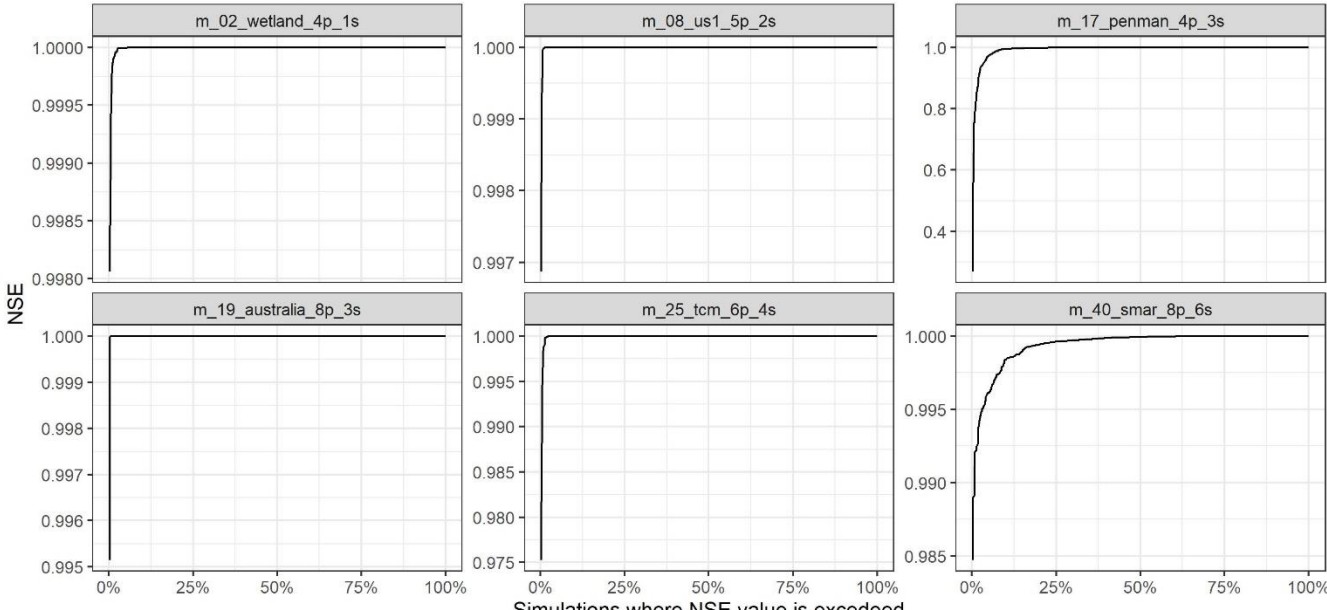

**Figure 4: NSE values calculated between streamflow simulated with MARRMoT v2.0 and v2.1. All models not shown had NSE>0.999 for all simulations.**


Finally, in Figure 5 we show the changes in runtime between the three versions used for this application test. Specifically, the plots show the ratios of runtime for models implemented in MARRMoT v2.1 (Figure 5a) and v2.0 (Figure 5b) to their runtimes in their MARRMoT v1.4 implementation. On average, models in their v2.1 implementaion ran 3.6 times faster than in their v1.4 implementation, however, improvements were on average higher for multi-store models with higher runtimes to start

with.

b shows that these improvements were generally due to the new root-finding routine rather than the object structure itself: despite there being some large differences across models, the object-oriented structure had on average nearly no impact on runtime (-1.2%). Nevertheless, the object structure was crucial for testing and deploying the new more efficient root-finding routine easily across all models.





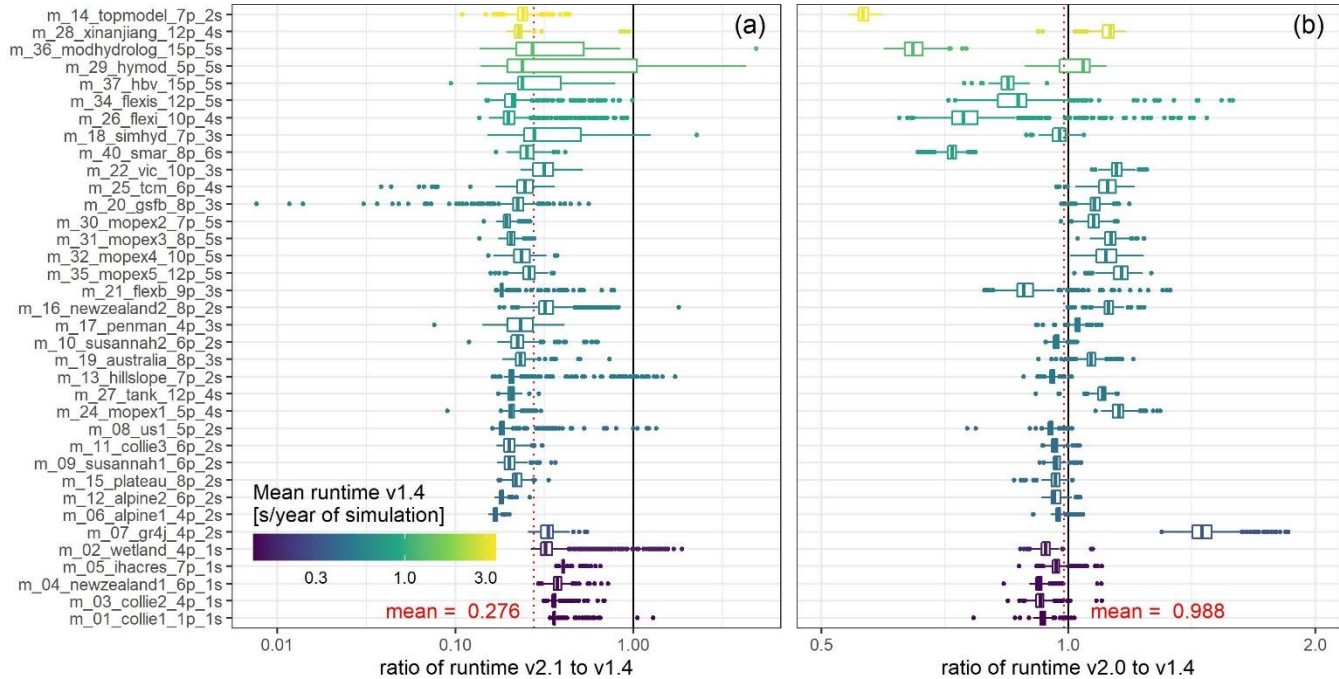

**Figure 5: Ratio of runtimes for models in MARRMoT v2.1 (a) and v2.0 (b) to their runtimes in version v1.4. Models are ordered by descending average runtime in v1.4 (see colour scale). Ratios < 1 indicate that the newer version is more computationally efficient than MARRMoT v1.4. Note that x-axes scales, albeit different, are logarithmic in both plots.**


## 4 Discussion

Amongst the other benefits highlighted in the previous sections, the object-oriented structure of MARRMoT v2 described here allowed for some additional insight into the hydrological models tested. In particular, the possibility to retrieve information about the quality of the ODEs' solution at every timestep highlighted that there are a handful of models in the framework

whose ODEs are particularly challenging to solve. Many of those models (namely *m_02_wetland*, *m_13_hillslope*, *m_28_xinanjiang* and *m_29_hymod*) share the flux process *saturation_2*, a nonlinear saturation excess from a store with different degrees of saturation, in their constitutive equations. While not all models containing this flux have instances where the framework cannot find a suitable solution (e.g. it is also contained in the equations of *m_22_vic*, whose equations are always solved satisfactorily within the test cases presented here), it is likely that the functional form of this flux definition

causes the resulting ODEs to be particularly challenging to approximate and solve. Note that MARRMoT v2.1 tries to solve the ODEs using three different solvers and several starting positions. While the new MARRMoT structure and root-finding routine used generally improved the ability of the framework to find acceptable solutions to these ODEs, this issue still persists in this version of the framework, however now it is known and actions can be taken in a future update to improve the





framework's ability to solve model equations in these cases. This could be achieved, for example, by implementing adapting
time-stepping schemes based on error estimates.

A similar approach could also help solve the significant discrepancies observed between equally valid solutions to the implicit
Euler approximation of the ODEs for the model *m_17_penman*, of which at most a single one corresponds to the actual solution
of the ODEs. This highlights the importance of separating the definition of models' constitutive equations to their
approximation and solution in order to better understand and reduce possible sources of errors and uncertainty. Specifically,
the application test described here suggests that the ODEs of *m_17_penman* may not be sufficiently constrained as they can
accept multiple equally valid solutions. Nevertheless, it is reassuring to observe that the discrepancies only occurred in a very
small percentage of the catchments tested, especially considering that the same or very similar equations to *m_17_penman* are
also contained in a handful of other models (e.g. *m_25_tcm*) that did not experience the same issues with this set of catchments.
This issue was not explored further for this MARRMoT release, but its better understanding and solution, through the
implementation in the framework of an appropriate error control mechanism, is expected to be prioritised for a future release.

## 5 Conclusions

In this paper, we presented a radical restructuring of the MARRMoT framework for hydrological modelling. The new
published version of the toolbox (v2.1) uses an object-oriented approach to represent model structures. Whereas the motivation
for the restructuring was to improve runtimes, we acknowledge that the improvements shown here, albeit significant, still fall
short of the speed-ups that may be expected from using languages such as Fortran or C. Nevertheless, as already mentioned
by Knoben et al. (2019) in their presentation of the toolbox, slower runtimes are the trade-off to accept for a toolbox that is
easy to use and understand and flexible enough to emulate a variety of model structures.

Overall, the version of MARRMoT presented here manages to find a balance between these competing objectives succeeding
to improve upon the previous versions of MARRMoT in terms of runtime and quality of the simulations, as well as readability
and ease of use. Additionally, it provides enhanced features to assess and debug errors and problems with model structures or
simulation parameters. Finally, it drastically simplifies the procedure for creating new and modifying existing models, by
providing a clear template for generating model class definitions as well as a fully developed superclass which already provides
most of the necessary code to run a model following the current best practices for model development. Ultimately, MARRMoT
offers accessible and shareable versioned code for many commonly used hydrological models. It provides an easy-to-use
framework for model calibration and simulation, model comparison and objective testing of modelling hypotheses.
Additionally, it allows hydrologists at all levels of academia and industry to experiment and play with model components and
equations within a well-designed modelling environment. We hope with this release to foster a culture of reproducible research,
code availability, curiosity and scrutiny towards our modelling tools and the ways they represent real hydrological systems
and eventually contribute to a deeper understanding of hydrological processes and the development of the next generation of
hydrological models.



**Code availability**

MARRMoT is provided under the terms of the GNU General Public License version 3.0. The MARRMoT v2.1 (Trotter and Knoben, 2022) code and user manual can be downloaded from https://github.com/wknoben/MARRMoT (last access: 12/05/2022, DOI: 10.5281/zenodo.6484372). MARRMoT has been developed on MATLAB version 9.11.0.1873467 (R2021b) and tested with Octave 6.4.0. To run in MATLAB, the Optimization Toolbox is required, while Octave requires the "optim" package. The user manual contains detailed description of the features of the framework and the models included; instructions and examples on how to run and calibrate models; and guidance on how to create new model structures, modify existing ones and contribute to the development of MARRMoT.

**Author contributions**

LT conceptualised and developed the new MARRMoT code, with technical support by WK and KF. KF, MS and MP provided supervision. LT wrote this article and WK, KF, MS and MP contributed to reviewing and editing.

**Acknowledgements**

We gratefully acknowledge the contributions of Philip Kraft, Sebastian Gnann, Clara Brandes, Koen Jansen, Mustafa Kemal Türkeri and Thomas Wöhling for various suggestions and improvements to MARRMoT v.1.1 to v1.4. Wouter Knoben was supported by the Global Water Futures program, University of Saskatchewan. Keirnan Fowler is supported by Linkage Project LP170100598, which is funded by the Australian Research Council, Victorian Department of Environment, Land, Water and Planning, the Bureau of Meteorology (Australia) and the Victorian Environmental Water Holder. Luca Trotter and Margarita Saft are supported by Linkage Project LP180100796, which is funded by the Australian Research Council, Victorian Department of Environment, Land, Water and Planning, and Melbourne Water.

**Competing interests**

The authors declare that they have no conflict of interest.

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
