# Peer review of "Modular Assessment of Rainfall–Runoff Models Toolbox (MARRMoT) v2.1: an object-oriented implementation of 47 of your favourite hydrologic models for improved speed and readability"

_Geoscientific Model Development, 2022_

## Author Response (AR1)

**Reviewer #1**

The manuscript here presents a new version of MARRMoT with significant changes in structure and solver. The new version implemented some good programming practices of modular, object-oriented structure to improve code readability and debugging as well as to reduce the repetitiveness of the code. Benchmarked again the previous version, the new structure and solver seem to be stable and consistent for most of the sub-models (30 out of 36). Overall, the improved speed and readability in the new version echo the efforts toward reproducible and transparent research and coding.

The manuscript is generally concise, organized, and well-written. I don't have major concerns about the manuscript but some minor comments that I hope to help the manuscript clarify a few points and potentially reach a broader audience and a larger number of users.

Thank you for taking the time to review our manuscript and comment on it. Please see our responses to specific points below, we have uploaded a revised version of our manuscript to integrate your comments and suggestions. Note that all line numbers indicated refer to the old version of the manuscript, unless otherwise indicated.

1, Line 205, for new users or those who are not familiar with the MARRMoT, what is the model spin-up process, what are the spin-up criteria (discharge, water storage / soil moisture), and does the model automatically handle the repeating of climate forcing data?

1. We added a reference in the revised version of the manuscript (l 214, revised version) pointing to the relevant section in the user manual. We also took the chance to include additional references to the user manual throughout Section 2 of the manuscript.

The reference to warm-up periods in line 205 relates to now being able to select the timesteps to calculate objective functions on, including during calibration. This can be used to set up warm-up periods ("using this function together with the calibration method allows to set up warm-up periods and specify what periods (even non-contiguously) to use to for calibrating the models." ll 206-207). The framework itself, however, does not provide spin-up criteria or automatic repeating of climate forcing.

2, does the model has the "hot-start" ability to continue running the model using saved model outputs from the last run?

**2.** This is an interesting suggestion that we will certainly consider for a future update. Currently this can be done, but does not come out-of-the-box as a feature in MARRMoT. A user would need to save the outputs of a simulation and use them to set up the starting states of a new simulation. We have added a new issue on the GitHub repository to mark this for future development.

3, it was not very clear whether the model comparison was made by comparing all NSE from 559 catchments or the median NSE? For example, does 0.29 mm/year the average difference for all 559 catchments?

**3.** In order to make sure the results of the comparisons we ran are clear, in the revised version of the manuscript, we have updated the captions to Figures 2 and 4 as well as flipped the axes of Figure 4 (in order to match in format the data presented in Fig 2). We do not believe it necessary to modify the text of Section 3.2 as all numbers in that section clearly indicate performance minima and are always preceded by specifiers such as "never higher than" (l 244), "never exceeded" (l 248), "at

least" (l 268) and "as low as" (l 270). The text explicitly mentions that the NSE scores in Figure 4 show that "**all** models have NSEs of at least 0.975 for **all** simulations" (l 268; emphasis added).

4, Figure 3, what does the label (subscript) mean, 5p_5s, 8p_3s, it might confuse readers.

**4.** We added a clarification on the meaning of MARRMoT models' names in Section 2.2 of the revised version. "All model classes retain the same naming convention as the model functions in MARRMoT v1, which includes a progressive identifier of the model within the framework, the general name of the model (replaced by the location of first application for unnamed models) and indicators of the numbers of parameters and stores of the model; for example the MARRMoT model class for GR4J (Perrin et al., 2003) is called *m_07_gr4j_4p_2s* indicating that it is the seventh model in the framework and has four parameters and two store." (ll 179-183, revised version)

5, Figure 5, the ratio is easy to show speed improvement compared to the previous version. But what were the computational time and the time difference compared to their original counterparts that wrote in C or Fortran?

**5.** We believe that comparison with non-MARRMoT versions of these models is outside the scope of this article. The scope of this paper is to present an update of MARRMoT from v1 to v2.1 and the comparisons focus on these versions of the models. Additionally, as mentioned in the original MARRMoT paper (Knoben et al., 2019), MARRMoT is based on existing model documentation, not code. It's practically impossible to track down the one true original code version of pretty much any of the models.

6, Line 300, I agree that adapting time-stepping schemes is critical and it might further improve speed and efficiency.

**6.** Thanks for the encouragement, we hope this improvement will be implemented soon.

**References:**

Knoben, W.J.M., Freer, J.E., Fowler, K.J.A., Peel, M.C., Woods, R.A., 2019. Modular Assessment of Rainfall-Runoff Models Toolbox (MARRMoT) v1.2: an open-source, extendable framework providing implementations of 46 conceptual hydrologic models as continuous state-space formulations. Geoscientific Model Development 12, 2463–2480. https://doi.org/10.5194/gmd-12-2463-2019

**Reviewer #2**

Thank you very much for your comments and suggestions. We have uploaded a revised version of our manuscript to address them. In our responses below we outline how for each specific points. Note that all line numbers in our response refer to the manuscript under discussion (old version), unless otherwise indicated.

Although I appreciate the quantitative comparision of v1.4 and v2.0 shown in Figure 2 it would be nice if there was an assessment of the conditions under which the largest differences occur (for example in m_34_flexis_12p_5s).

As mentioned in the manuscript under discussion (ll244-247), the largest differences occur in the models where the stores ODEs are solved differently between v1.4 (sequentially) and 2.0 (concurrently). We believe this is responsible for the largest discrepancies as it "may introduce errors in v1.4 that are not present when all stores are solved simultaneously in v2.0." (ll 246-247)

Section 3.1 - Although Knoben et al. (2020) details the calibration process used it would be helpful to summarize the process used in this paper.

Regarding the calibration process described by Knoben et al. (2020), we included some additional context Section 3.1 of the revised version. "The authors calibrated the models using the Covariance Matrix Adaptation Evolution Strategy (CMA-ES) algorithm (Hansen and Ostermeier, 1996; Hansen et al., 2003) to optimise the Kling-Gupta Efficiency (KGE, Gupta et al., 2009). The parameter values they found are available as supplementary material to Knoben et al. (2020)." (ll 231-234, revised version)

Line 300 - Add reference for "implementing adapting time-stepping schemes based on error estimates (ref)"

Finally, thanks for pointing out the lack of reference in line 300. We added a reference to (Clark and Kavetski, 2010) in the revised version of the manuscript (l 309, revised version).

**References**:

Knoben, W.J.M., Freer, J.E., Peel, M.C., Fowler, K.J.A., Woods, R.A., 2020. A Brief Analysis of Conceptual Model Structure Uncertainty Using 36 Models and 559 Catchments. Water Resources Research 56, 1–23. https://doi.org/10.1029/2019WR025975

Clark, M.P., Kavetski, D., 2010. Ancient numerical daemons of conceptual hydrological modeling: 1. Fidelity and efficiency of time stepping schemes. Water Resources Research 46. https://doi.org/10.1029/2009WR008894

---

## Editor Decision (ED1)

Geosci. Model Dev. Discuss., referee comment RC1
https://doi.org/10.5194/gmd-2022-135-RC1, 2022
**Comment on gmd-2022-135**

Anonymous Referee #1

Referee comment on "Modular Assessment of Rainfall–Runoff Models Toolbox (MARRMoT) v2.1: an object-oriented implementation of 47 of your favourite hydrologic models for improved speed and readability" by Luca Trotter et al., Geosci. Model Dev. Discuss., https://doi.org/10.5194/gmd-2022-135-RC1, 2022

The manuscript here presents a new version of MARRMoT with significant changes in structure and solver. The new version implemented some good programming practices of modular, object-oriented structure to improve code readability and debugging as well as to reduce the repetitiveness of the code. Benchmarked again the previous version, the new structure and solver seem to be stable and consistent for most of the sub-models (30 out of 36). Overall, the improved speed and readability in the new version echo the efforts toward reproducible and transparent research and coding.

The manuscript is generally concise, organized, and well-written. I don't have major concerns about the manuscript but some minor comments that I hope to help the manuscript clarify a few points and potentially reach a broader audience and a larger number of users.

1, Line 205, for new users or those who are not familiar with the MARRMoT, what is the model spin-up process, what are the spin-up criteria (discharge, water storage / soil moisture), and does the model automatically handle the repeating of climate forcing data?

2, does the model has the "hot-start" ability to continue running the model using saved

model outputs from the last run?

3, it was not very clear whether the model comparison was made by comparing all NSE from 559 catchments or the median NSE? For example, does 0.29 mm/year the average difference for all 559 catchments?

4, Figure 3, what does the label (subscript) mean, 5p_5s, 8p_3s, it might confuse readers.

5, Figure 5, the ratio is easy to show speed improvement compared to the previous version. But what were the computational time and the time difference compared to their original counterparts that wrote in C or Fortran?

6, Line 300, I agree that adapting time-stepping schemes is critical and it might further improve speed and efficiency.

[Figure]

Geosci. Model Dev. Discuss., referee comment RC2
https://doi.org/10.5194/gmd-2022-135-RC2, 2022
**Comment on gmd-2022-135**

Anonymous Referee #2

Referee comment on "Modular Assessment of Rainfall–Runoff Models Toolbox (MARRMoT) v2.1: an object-oriented implementation of 47 of your favourite hydrologic models for improved speed and readability" by Luca Trotter et al., Geosci. Model Dev. Discuss., https://doi.org/10.5194/gmd-2022-135-RC2, 2022

Although I appreciate the quantitative comparision of v1.4 and v2.0 shown in Figure 2 it would be nice if there was an assessment of the conditions under which the largest differences occur (for example in m_34_flexis_12p_5s).

Section 3.1 - Although Knoben et al. (2020) details the calibration process used it would be helpful to summarize the process used in this paper.

Line 300 - Add reference for "implementing adapting time-stepping schemes based on error estimates (ref)"